# Crystal Structure and Properties of Thallium(I) Salinomycinate

**DOI:** 10.3390/ijms26136504

**Published:** 2025-07-06

**Authors:** Nikolay Petkov, Petar Dorkov, Angel Ugrinov, Elzhana Encheva, Miroslav Abrashev, Diana Zasheva, Teodora Daneva, Ivayla N. Pantcheva

**Affiliations:** 1Faculty of Chemistry and Pharmacy, Sofia University “St. Kliment Ohridski”, 1164 Sofia, Bulgaria; elzhanae@uni-sofia.bg; 2Research and Development Department, Biovet Ltd., 4550 Peshtera, Bulgaria; p_dorkov@biovet.com; 3Department of Chemistry and Biochemistry, North Dakota State University, Fargo, ND 58108, USA; angel.ugrinov@ndsu.edu; 4Faculty of Physics, Sofia University “St. Kliment Ohridski”, 1164 Sofia, Bulgaria; mvabr@phys.uni-sofia.bg; 5Institute of Biology and Immunology of Reproduction, Bulgarian Academy of Sciences, 1113 Sofia, Bulgaria; zasheva.diana@yahoo.com (D.Z.); danevadoki@abv.bg (T.D.); 6Centre of Competence “Sustainable Utilization of Bioresources and Waste of Medicinal and Aromatic Plants for Innovative Bioactive Products” (BIORESOURCES BG), 1164 Sofia, Bulgaria

**Keywords:** polyether ionophore, mononuclear Tl(I) complex, crystal structure, antimicrobial activity, antitumor properties

## Abstract

In this study, we present the preparation and characterization of a novel thallium(I) coordination compound of the polyether ionophorous antibiotic salinomycin (SalH). The complex [TlSal(H_2_O)] exists as two subunits, SalTl1 and SalTl2, which differ slightly in their structural parameters. Salinomycin acts in a pentadentate coordination mode through oxygen donor atoms, and the six-fold arrangement around the metal centers is completed by interaction with a water molecule. In the overall complex structure, the two mononuclear species SalTl1 and SalTl2 are connected via a hydrogen bond network by a third water molecule. The inclusion of the heavy metal ion into the structure of the polyether ionophore reduces its biological activity against Gram-positive microorganisms and cervical cancer cells at in vitro conditions.

## 1. Introduction

Salinomycinic acid (SalH, Figure 1) is a veterinary drug used for prevention of coccidiosis in the stock farming. It belongs to the family of polyether ionophores (PIs)—natural biologically active compounds produced by strains of *Actinomycetota* phylum (*Streptomyces* or *Actinomyces* genera). The clinically approved PIs include monensin (*Streptomyces cinnamonensins*) [1,2], salinomycin (*Streptomyces albus*) [3,4], lasalocid (*Streptomyces lasaliensis*) [5,6], narasin (*Streptomyces aureofaciens*) [7,8], maduramicin (*Actinomadura yumaensis*) [9,10], laidlomycin (*Streptomyces* spp.), and semduramicin (*Actinomadura roserufa*) [11], among over 120 polyether ionophores known today [12,13,14]. In addition to their coccidiostatic properties, these compounds are also used in practice as antibiotics against infections caused by Gram-positive bacteria.

PIs are polyether derivatives of monocarboxylic acids and consist of multiple oxygen-containing heterocyclic residues linked as spiroketal fragments [15,16]. The antibiotics exist generally in a pseudo-cyclic form due to the “head-to-tail” cyclization, with most polar oxygens directed inward and with nonpolar alkyl substituents oriented outward. The formed hydrophylic oxygen-rich cavity is able to accommodate water or monovalent metal ions [1,4,17,18,19,20,21,22,23,24,25,26]. For that reason, most PIs possess the ability to coordinate alkali metal cations as neutral complexes and are known as monovalent ion binders. However, the length of the polyether chain is responsible for the selectivity of PIs towards alkali ions (generally Na^+^ vs. K^+^), which varies depending on ionophore structure. The monovalent PIs form lipid-soluble metal complexes with group 1 metal cations, and the commercially available veterinary products contain the corresponding sodium coordination species. The neutral complexes easily penetrate through the cell membranes of sporozoites and bacteria, increasing the intracellular concentration of Na^+^ ions. The subsequent disturbance of metal homeostasis activates numerous energy-dependent processes, which result in cell death [27,28].

The binding affinity of monovalent PIs to the essential alkali cations can be further transferred to their ability to complex heavier metal(I) cations as Cs^+^ or Tl^+^, and, in such a way, to serve as potential antidotes in the case of acute/chronic poisoning that may occur in livestock farming [29,30]. At present, we have focused our study on the disposal of thallium(I), which is a nonessential element and is more toxic to living organisms than mercury, lead, and cadmium. In our study, we found that the salinomycinate anion (Sal^−^) binds Tl^+^ ion as a mononuclear coordination species (SalTl). The structure of SalTl was solved by Single-Crystal X-ray Diffraction (SCXRD), and the complex was characterized by Infrared (IR) and Raman spectroscopies, mass spectrometry, and elemental analysis. To the best of our knowledge, the coordination species described here are the second example of a monovalent metal derivative of the natural unmodified salinomycin, in addition to its sodium analogue [4]. Furthermore, we studied the effect of thallium(I) binding on antibacterial and antiproliferative properties of salinomycin using a set of Gram-positive bacteria and the human tumor HeLa cell line (cervical cancer).

## 2. Results and Discussion

The interaction of salinomycinate anion (Sal^−^, Figure 1) with thallium(I) nitrate in a mixed solvent system leads to the spontaneous formation of mononuclear coordination species [TlSal(H_2_O)] (called SalTl). The total composition of the complex can be described as [TlSal(H_2_O)]×0.5H_2_O due to the binding of water molecules with two SalTl subunits via hydrogen bonding. Its Time-of-Flight High-Resolution Mass Spectrum (ToF HRMS) in a positive mode (ESI+, Electro Spray Ionization) is dominated by peaks assigned to fragments containing thallium salinomycinate [SalTl]H^+^ (obs. 955.4661 *m*/*z*, mass error (m.e.) 0.11 ppm), sodium salinomycinate [SalNa]H^+^ (obs. 773.4819 *m*/*z*, m.e. 0.39 ppm), and Tl^+^ (obs. 204.9760 *m*/*z*, m.e. 7.81 ppm).

### 2.1. Description of the Crystal Structure of Thallium(I) Salinomycinate

The mononuclear SalTl complex crystallizes in space group P_1_. The asymmetric unit comprises two thallium(I) salinomycinates and three water molecules (Figure 1). The conformation of the two coordination species (referred to as SalTl1 and SalTl2) is practically identical, with slight differences in some bond lengths and angles (Table 1). Each metal center is captured by one antibiotic monoanion through five donor atoms from bidentate carboxylate (O1, O2) and carbonyl (O5) groups, furan (O9), and pyran (O10) moieties. The sixth place in the coordination sphere of Tl^+^ ions is occupied by water molecules—O1S in SalTl1 and O3S in SalTl2. The distances between thallim(I) ions and the donor atoms in both conformers vary from 2.70 to 3.33 Å and corroborate well the bond-length distribution of Tl^+^ cations bonded to oxygen [31].

The six-fold arrangement around thallium(I) ions in the two conformers can be described as a heavily distorted octahedron. The oxygens O1, O5, O10, and O1S (SalTl1) or O3S (SalTl2) and the metal center form planes with root-mean-square deviations (RMSD) of 0.077Å for Tl1 and 0.067Å for Tl2 (Figure 2). The donor atoms O2 and O9 complete the inner coordination shell, being far apart from the plane as follows: O2—1.490(8) Å (SalTl1), 1.573(8) Å (SalTl2); O9—2.268(6) Å (SalTl1), 2.246(6) Å (SalTl2). The O2−Tl−O9 angles are 119.3(1)° (SalTl1) and 125.1(2)° (SalTl2), respectively.

In the individual structure of each SalTl conformer, the salinomycinate exists in a “closed” form due to its “head-to-tail” folding. The hydroxyl groups O4H, O8H, and O11H do not bind to the metal(I) ions, but participate in the formation of intra- and intermolecular H-bond network, which additionally stabilizes the complex units and their packing. Four internal hydrogen bonds in SalTl1 and SalTl2 are realized by the interaction of O4H with O1, and O2/O8H/O11H with O1S (SalTl1) or O3S (SalTl2). The two conformers are linked by the O11_1–O8_2 intermolecular H-bonding of 2.703 Å distance, and the water molecule O2S participates in interactions with O8H (SalTl1) and O2/O11H (SalTl2), respectively (Table 2). The crystal packing is presented in Appendix A.

The existence of SalTl conformers resembles the case of sodium salinomycinate (SalNa), where the presence of two different complex species was observed (GIZDIC [4]). In contrast to the thallium complex reported here, Na^+^ ions are 5- (SalNa1) or 6- (SalNa2), coordinated with the tetradentate salinomycinate anion and one (SalNa1) or two (SalNa2) water molecules, ligated to the central metal ion. The carboxylate group acts in a monodentate manner, and similarly to SalTl, the hydroxyl groups are not directly involved in an interaction with sodium cations. In the sodium complexes of 20-deoxy (ESUZEX [21]) and 20-oxo (BIDWER [22]) derivatives of salinomycin, the metal ions are 5-coordinated, with a bidentate “head” carboxylate group in the first coordination species and a “tail” hydroxyl group in the latter, without water molecules as ligands. These data highlight the role of X-ray diffraction analysis as a crucial method for determining the exact structure of complex compounds in the solid state and the conformation of the organic binder there.

The same is also true for the known thallium(I) complexes of other polyether ionophores, where the antibiotics act differently. Thus, the shortest representative of PIs, lasalocid, forms monomer and polymer species with Tl^+^ cations, serving as a pentadentate ligand (JUFFOF, JUFFUL [32], JUFFOF01 [33]). Grisorixin is bound to the metal center via five O-donor atoms (GRISTL [34]), while cationomycin (BINYUS [35]) and emericid (EMERTL [36]) act as hepta- and octadentate binders, respectively. In all these complexes, no water molecules are bound to the metal(I) ion, and except emericid, the carboxylate function is coordinated in a monodentate way. The Tl−O distances range from 2.60 to 3.07 Å and are in line with the data observed for the newly synthesized SalTl complex.

In addition to the SCXRD studies, we also carried out a powder XRD experiment using the bulky sample of SalTl (Figure 3). The obtained results corroborate very well the calculated pattern from the crystal structure, confirming the content and purity of the entire material isolated from the crude reaction mixture. Combining the present data with those from elemental analysis (seeSection 3) and mass-spectrometry, we conclude that the bulky specimen of SalTl structurally resembles the one solved by SCXRD.

### 2.2. Spectral Characterization of SalTl

The IR spectrum of salinomycinic acid (SalH, Figure 4, Table 3) exhibits two specific bands that undergo a change upon complexation with thallium(I) ions. The first one appears as an asymmetric signal at 1709 cm^−1^ with a half-width of 40 cm^−1^. It is assigned to the stretching vibrations of the carboxylic (1716 cm^−1^) and the H-bonded carbonyl (1704 cm^−1^) groups of the ligand. The second absorbance band is observed in the range of 3680-3340 cm^−1^ (centered at ca. 3505 cm^−1^; half-width of 162 cm^−1^) and is attributed to the OH-stretching vibrations within the SalH structure.

The IR spectrum of SalTl complex (Figure 4) shows some significant differences compared to that of SalH. It contains the following:-A symmetric band at 1703 cm^−1^, which is narrower (half-width of 28 cm^−1^) and less intense due to the carbonyl stretching of the isolated C=O group, engaged in Tl–O interaction according to the X-ray findings; -Two new bands at 1549 cm^−1^ and 1398 cm^−1^ (not observable in the spectrum of SalH), which are assigned to the asymmetric and symmetric stretches of the carboxylate function. These signals evidence the deprotonation of the carboxylic group in SalH under complexation with Tl^+^ cations. Furthermore, the magnitude of Δ (the difference between the two vibrations), calculated as 151 cm^−1^, is similar but still lower than that of the corresponding ionic values, and can be used as an indicator of an asymmetric bidentate coordination mode of the carboxylate function [37]. In contrast, this difference is 161 cm^−1^ for SalNa, where the carboxylate anion acts in a monodentate fashion [4], and is 159 cm^−1^ for SalK, supposed to be isostructural with SalNa;-An asymmetric broad band in the range of 3680–3120 cm^−1^ (centered at ca. 3390 cm^−1^, half-width of 312 cm^−1^) due to the OH-stretching vibrations of hydroxylic groups of salinomycinate ligand and water molecules in SalTl, engaged in the formation of Tl−O and H-bonds.

In addition to the IR study, the same set of compounds was explored by Raman spectroscopy (Figure 5) as a complementary technique to more precisely discuss some important features of SalH and its complexes with monovalent cations of Na, K, and Tl. First, we were able to clearly confirm the absence of crystallinity for two of the samples, evident from the broader signals observed in the Raman spectra of SalH (black) and SalK (red) due to their amorphous character. In contrast, SalNa (blue) and SalTl (teal) were examined as monocrystals, apparent from the well-resolved bands in the corresponding spectra. As confirmation of the crystallinity of the last two samples, the results of their study with polarized light can be considered as an additional point. The observed change in the intensity of the individual bands in the Raman spectra (Appendix A) proves that the analyzed samples of SalNa and SalTl are crystalline, and the crystals are larger than the spot size of the laser beam. Such behavior of the bands in the spectra of the studied specimens of SalH and SalK is not detected, from which their amorphous character can be deduced.

Secondly, the Raman spectroscopy allowed us to unambiguously assign the stretching vibrations of the isolated C=O and C=C bonds in SalH, which, due to the higher polarizability, appear as intense signals compared to the IR spectra. Thus, we confirmed the position of the isolated C=O vibration stretch of the salinomycinic acid (Table 3) but also assigned the absorbance of its C=C vibration.

In the third place, the Raman spectra show changes in the band positions due to the presence of a double bond in the structure of the salinomycinate ligand. In three of the cases, the band is observed as a singlet (SalH, SalK, SalTl), while in the sodium complex it is registered as a doublet arising from the Davydov splitting explained below. Looking at the spectra of three complexes, there is a clear trend that the Raman shift of the C18=C19 band is proportional to the distance of the metal ion to it. Two possible explanations can be given here. On one side, the volume of the ion, which directly affects the cavity size it requires, leads to structural differences in the complexes studied. The conformation change impacts the strain in the cyclic systems, with the cyclohexenol fragment showing a difference in the value of valence angles between SalNa and SalTl coordination species (Table 4). The comparable data for SalH and SalK can be explained by the fact that salinomycin is a potassium ionophore [38], and the ligand most likely forms a cavity of similar dimensions in its free and bound forms. An alternative explanation can be given by the different electronic configuration of the monovalent metal ions, which has been shown to affect the vibrational characteristics in unsaturated systems [39]. This can be confirmed by the position occupied by the metal ion in the structure of the complexes, which is perpendicular to the double bond, allowing direct interaction with its p-orbitals.

Fourth, but not least, the Raman experiment is advanced, showing the clear difference between SalNa and SalTl crystals, which reflects the participation of the ligand in their inner coordination sphere according to SCXRD analysis. Thus, a strong Davydov splitting is observed in the SalNa spectrum (blue) in the range of 1750–1650 cm^−1^, much less detectable in the corresponding spectrum of SalTl (teal). This fact is attributed to the presence of two different conformers of the salinomycinate ligand in the unit cell of SalNa [4], while the antibiotic anions are coordinated in a very close manner in the corresponding SalTl1 and SalTl2 complex units.

### 2.3. Biological Effect of SalTl and Related Compounds

The activity of SalTl on the growth of aerobic Gram-positive microorganisms was evaluated by the double-layer agar diffusion assay. The choice of the laboratory strains is based on the infections caused by the bacteria against which the properties of potential drug candidates are to be tested:-*Bacillus subtilis* (BS) is considered non-pathogenic to humans; however, it causes infections of the respiratory and urinary tracts, of dermal wounds and burns, as well as secondary nosocomial infections in immunocompromised patients [40,41];-*Bacillus cereus* (BC) infections are associated with two types of food-borne gastroenteritis: emetic illness (due to the produced very stable toxin that is resistant to high temperatures and pH changes) and diarrhea (mediated by a less stable enterotoxin complex) [42];-*Kocuria rhizophila* (KR) is an outstanding example of an emerging fish pathogen [43,44], although in rare cases, it can be involved in human infections [45];-*Staphylococcus aureus* (SA) is an extraordinarily versatile pathogen responsible for staphylococcal food poisoning, hospital and community-acquired infections, as well as toxic shock syndrome [46];-*Staphylococcus saprophyticus* (SS) causes several rare infections, such as pyelonephritis, infective endocarditis, meningitis, as well as urinary tract infection, endophthalmitis, and uncomplicated cystitis [47].

For the sake of comparison, salinomycinic acid (SalH), monensic acid (MonH), the commercially available sodium salinomycinate (SalNa) and sodium monensinate (MonNa), as well as thallium(I) monensinate (MonTl) and TlNO_3_, were also included in the tests. The determined minimum inhibitory concentration (MIC, µM) of the compounds assesses their ability to inhibit the visible growth of the bacterial strains. It should be borne in mind that the lower the MIC value, the better the antibacterial agent is.

Monovalent thallium ion is similar to the potassium (K^+^) in ionic radius and electrical charge, mimicking its properties, which contributes to the toxic nature of Tl^+^ in vivo [48] while the studies on the activity of TlNO_3_ against various bacterial strains reveal its low in vitro antimicrobial potential. Our data also show that thallium(I) nitrate does not inhibit the visible growth of the tested microorganisms at the highest concentration applied (1 mg/mL, 3.8 mM), except the strain of *K. rhizophila* (MIC 1.87 mM).

The antibacterial assay regarding the efficacy of ionophores and their metal derivatives outlines the following trends (Figure 6):(1)The strain of *B. cereus* (orange boxes) is the most sensitive to the effects of the tested compounds, while *K. rhizophila* (green boxes) seems to be resistant except to SalNa and SalH;(2)Sodium salinomycinate (SalNa) appears to be the most toxic against the entire bacterial set, with its activity varying within a relatively narrow MIC range of 2.5–40.4 µM;(3)Thallium salinomycinate (SalTl) possesses the lowest toxicity and is less active than the corresponding monensinate counterpart (MonTl).

Antibiotics and corresponding metal derivatives can be hierarchically ranked according to their decreasing inhibitory effect as follows:BS: SalNa ≈ MonH > SalH > MonTl > MonNa > SalTlBC: SalNa ≈ MonNa > MonTl > SalH > MonH > SalTlKR: SalNa > SalH > MonH > MonTl > MonNa > SalTlSA: SalNa ≈ MonTl ≈ MonNa ≈ MonH > SalH > SalTlSS: SalNa ≈ MonTl > SalH ≈ MonNa ≈ MonH > SalTl.

The in vitro data obtained cannot be simply explained, as they show ambivalent trends in the effect of thallium(I) ions on the antibacterial activity of polyether ionophores. Most likely, the binding of Tl^+^ to salinomycin leads to the formation of sufficiently stable coordination species that are unable to dissociate in the intracellular space of microorganisms. Thus, neither the antibiotic nor the toxic metal ion can exert its properties individually or synergistically. This is apparently not the case with thallium(I) monensinate, which expresses more pronounced activity, although variable across the bacterial strains tested. However, the observed differences in the susceptibility of microorganisms to structurally similar compounds indicate that each therapeutic candidate should be subjected to the most complete possible investigation of its biologically relevant properties.

To further evaluate the effect of Tl(I) ions on the biological performance of salinomycin and monensin, we also conducted a pilot study using cervical cancer cells (HeLa). The HeLa cells were selected since they belong to one of the four female cancers diagnosed globally [49]. Unfortunately, the survival rate of patients using conventional surgery and chemotherapy is still low [50], prompting a search for new drugs with more efficacy.

The cytotoxic concentration 50 (CC_50_, µM) of the compounds, which decreases the viability of the cells by 50% on the 24^th^ hour after the treatment, was estimated by the crystal violet (CV) assay [51]. The target ionophores, their sodium and thallium complexes, and TlNO_3_ were applied as stock solutions in DMSO (1 mg/mL) reaching the final concentrations of 25, 10, 5, 1 and 0.5 µg/mL in working solutions using Dulbecco’s Modified Eagle’s Medium (DMEM) supplemented with 5% fetal bovine serum (FBS) as a dilutant. The effect of DMSO (2.5−0.05% in the respective solutions) on the proliferation of HeLa cells was found to be insignificant.

The experimental data show that the toxicity of the compounds is concentration-dependent (Figure 7), with thallium salt again being the least toxic among the studied set. The CC_50_ values of MonH and MonTl were directly determined from the dose-effect curves. The data for the others were estimated from the trendline built from the response range where a linear relationship was observed. As evident, the toxicity of SalH, SalNa, and SalTl is very similar, while the compounds within the monensin series can be ordered with respect to their increasing activity as MonNa < MonH < MonTl. The diverse results observed at this stage of the modelling study appear to be due to complicated interrelationships, among which (as in the case of microbial assay), the different stability of the compounds in the specific culture medium, and therefore their different dissociation degrees, are important. Based on the obtained data, we refrain from speculating on the mechanisms by which salinomycin, monensin, and their metal derivatives reduce HeLa cell viability. These effects most likely relate to the facts that (i) salinomycin inhibits the growth of breast cancer (which is reproductive like cervical cancer and exhibits the same signalling pathway) [52] and (ii) monensin affects various signaling pathways, including the insulin growth factor 1 receptor, essential for the development of reproductive cancer [53].

On the other hand, the observed low ability of the pure thallium(I) salt TlNO_3_ to inhibit in vitro growth of microorganisms and cervical cancer cells can probably be explained by (i) its ionic nature and inefficient penetration through cell membranes into the intracellular space and (ii) the possibility of “blocking” negative effects by the protein components of the culture medium limiting its cellular uptake.

To obtain more detailed information about the changes in cell morphology upon treatment with the compounds studied, May–Grünwald Giemsa (Pappenheim) staining was also performed (Figure 8). In the control group, the cells retain their normal morphology and dimensions, and no alterations in nuclear chromatin were observed. In contrast, the treatment with ionophores and their metal complexes induces remarkable changes: most cells became rounded, although the overall size is largely preserved. In cells treated with SalH, SalNa, and TlNO_3_, the cell shape remains intact in some areas. All specimens exposed to ionophores, their metal complexes, and thallium nitrate (except SalTl) exhibit karyopyknosis, with particularly pronounced chromatin condensation observed in cells treated with SalH and MonTl. Vacuolization, a marker of cell death, is also present in some cells, especially in those exposed to MonH, its derivatives, and SalNa. Notably, SalTl-treated cells preserve their structure and size, which can be explicated by the stable complex formation and its inability to dissociate intracellularly. The observed unaffected cell morphology agrees with the low toxicity of SalTl against HeLa (CV assay) but can also be indirectly used to explain the data obtained from the antimicrobial studies.

In summary, we used several Gram-positive microorganisms and the adherent cervical cancer cell line HeLa as model systems to evaluate the in vitro toxicity of monensin, salinomycin, and their sodium/thallium coordination species. On the one hand, the binding of thallium ions to salinomycin reduces its bioavailability and, accordingly, its effectiveness. So, this complex is not useful as an antimicrobial and/or anticancer agent. On the other hand, the stability of SalTl discloses the potential of salinomycin for its application as a detoxifier in poisoning with thallium compounds. To confirm these initial results, additional studies are needed regarding the properties of the tested systems in solution, as well as using in vivo models, which are beyond the scope of the present study.

## 3. Materials and Methods

### 3.1. Reagents and Materials

Sodium salinomycinate (SalNa, p.a.) was provided by Biovet Ltd. (Huvepharma, Peshtera, Bulgaria); HCl, KOH, Et_4_NOH (40% in H_2_O) were purchased from Sigma-Aldrich (St. Louis, MO, USA), and TlNO_3_, acetonitrile (MeCN), methanol (MeOH), diethyl ether (p.a. grade) were delivered from local suppliers.

The sterile plastic ware was supplied by local companies. Distilled water was used when required.

### 3.2. Synthesis of SalTl

A solution of TlNO_3_ (0.2 mmol, 53.3 mg in 5 mL MeCN) was added, dropwise at constant stirring, to a solution containing SalH (0.1 mmol, 75.1 mg in 10 mL MeOH) and 0.1 mmol Et_4_NOH (36 µL, 40% in H_2_O). An equal volume of water (15 mL) was added to the resulting solution and allowed to evaporate slowly at room temperature. The resulting transparent colorless crystals from SalTl were washed repeatedly with water.

SalTl: [Tl(C_42_H_69_O_11_)(H_2_O)]×0.5H_2_O, MW 981.41 g/mol. Elem Anal.: Calc. H, 7.40; C, 51.40%. Found: H, 7.09; C, 50.10%. Yield: 87.3 mg (88%).

### 3.3. X-Ray Crystallography

The crystals were mounted on a Bruker APEX-II CCD diffractometer (Billerica, MA, USA) for data collection at 107 K. The preliminary set of cell constants was calculated from the reflections collected from four sets of 30 frames, which produced the initial orientation matrices. Data collection was carried out using IµS Cu Kα radiation with a detector distance of 4.0 cm. The randomly oriented region of reciprocal space was surveyed to the extent of one sphere and a resolution of 0.84 Å. Total of 41 ω- and φ-scan sections of frames with 1.0° width were collected to achieve the desired completeness of 99.5%. The intensity data were corrected for absorption and decay [54]. The final cell constants were calculated from the xyz-centroids of the strongest reflections from the actual data collection after integration [55].

The structures were solved and refined by the SHELX [56] set of programs with Olex 2 v.1.5 software package [57]. The SHELXT [58] program was used to provide most of the non-hydrogen atoms, and the full-matrix least squares/difference Fourier cycles were performed to locate the remaining non-hydrogen atoms. All non-hydrogen atoms were refined with anisotropic displacement parameters. All hydrogen atoms were placed in ideal positions and refined as riding atoms with relative isotropic displacement parameters. The data collection parameters and refinement information for the single-crystal X-ray diffraction experiments are summarized in Table 5. Images were generated using CrystalMaker Software Ltd., Oxford, England (www.crystalmaker.com (accessed on 12 March 2025)).

### 3.4. Physical Measurements

The powder diffraction pattern was obtained at room temperature on a PANalytical Empyrean X-ray powder diffractometer (Malvern Panalytical, Malvern, UK) with Cu Kα radiation (λ = 1.5418 Å) operating at 40 kV, 30 mA.

IR studies in KBr pellets were performed on a Nicolet 6700 FT-IR, Thermo Scientific (Madison, WI, USA). Raman spectra were recorded on a LabRAM HR Visible Raman spectrometer in backscattering configuration (HORIBA Jobin Yvon, Kyoto, Japan). A 633 nm He-Ne laser line was used for excitation. An x50 objective was applied both to focus the incident laser light onto the sample surface and to collect the scattered light. The polarization direction of the incident linearly polarized laser light was changed using a lambda/2 plate. The scattered light was analyzed with a polarizer. To avoid possible local overheating of the laser, the laser power on the laser spot (with a diameter of about 2 µm) on the sample surface was reduced to 0.5 mW.

ToF HRMS measurements were carried out on a Waters SYNAPT G2-Si Q-ToF instrument (Waters Corporation, Milford, MA, USA). The samples were dissolved in methanol and directly injected for mass spectrometry analysis in a positive ionization mode. ESI conditions were as follows: capillary potential 3.0 kV (ESI+), sample cone potential 40 V, temperature source 90 °C, desolvation temperature 250 °C, desolvation gas flow 350 L/h. The observed range was set from 50 to 2000 *m*/*z*.

The C, H analysis was performed on a Thermo Fisher Flashsmart CHN/O (Thermo Fisher Scientific, Waltham, MA, USA).

### 3.5. Antibacterial Activity

The Gram-positive bacteria in the present study include the strains of *Bacillus subtilis* (BS, NBIMCC 1050, ATCC 11774), *Bacillus cereus* (BC, NBIMCC 1085, ATCC 11778), *Kocuria rhizophila* (KR, NBIMCC 159, ATCC 9341), *Staphylococcus aureus* (SA, NBIMCC 509, ATCC 6538), and *Staphylococcus saprophyticus* (SS, NBIMCC 3348). Nutrient agar (pH 7.2–7.4) containing meat extract (3 g/L) and peptone (5 g/L) and bacterial strains were purchased from the National Bank for Industrial Microorganisms and Cell Cultures (NBIMCC, Sofia, Bulgaria).

The minimum inhibitory concentration (MIC) of compounds causing the visible inhibition of the strain’s growth was determined using the double-layer agar diffusion method. Briefly, sterile (10 mL) and inoculated (10 mL, 1.5% inoculum, McFarland 4, and A_650_ = 0.8−1) agar were sequentially poured onto Petri dishes (90 mm). After agar solidification, holes of 6 mm diameter were punched and filled with 20 µL of the target compounds dissolved in MeOH. The test solutions were obtained by subsequent two-fold dilution of the stocks (1 mg/mL) up to 0.25 µg/mL. The diameter of the inhibited zones was measured after incubation at 30 °C for 24 h. Totally, nine readings from three separate experiments were collected. Methanol served as a negative control. The MIC values were recalculated in µM units, considering the molar mass of the tested compounds.

### 3.6. Cell Cultivation and Cytotoxicity Assay

The permanent cell line established from uterine cervix (HeLa, ATCC cell culture collection N CCL-2) was used as a model system in the present study. The cells were grown as monolayer cultures in Dulbecco’s Modified Eagle’s Medium (DMEM), supplemented with 10% fetal bovine serum (FBS), 100 U/mL penicillin, and 100 μg/mL streptomycin. The culture was maintained at 37 °C in a humidified CO_2_ incubator (5%). The adherent cells were detached using a mixture of 0.05% trypsin and 0.02% EDTA. The experiments were performed during the exponential phase of cell growth. The number of vital cells was counted hemocytometrically using the trypan blue exclusion method [60].

The stock solutions of the tested compounds (1 mg/mL) were prepared in DMSO. The working solutions (25, 10, 5, 1, 0.5 μg/mL) were obtained by subsequent dilution with DMEM, supplemented with 5% FBS, 100 U/mL penicillin, and 100 μg/mL streptomycin.

The cells were seeded in 96-well flat-bottomed microplates at a concentration of 1 × 10^4^ cells per well. After growth for 24 h to a subconfluent state (~60–70%), the monolayers were washed with phosphate-buffered saline (PBS, pH 7.2) and covered with 100 µL working solution. Each solution was applied to 4 wells. Cells grown in non-modified DMEM served as controls. After 24 h of incubation, the effect of the compounds on cell viability and proliferation was examined by crystal violet (CV) staining assay [51]. After each well was washed with PBS, the cells were fixed for 20 min and stained with 0.02% CV solution in methanol (100 µL) for 40 min. The excess dye was washed, and the wells were filled with 100 µL 70% ethanol (10 min) to extract the bound CV from the cells. Optical density was measured at 590 nm using an automatic microplate reader (SPECTROstar Nano, BNG Labtech, Ortenberg, Germany). Relative cell viability, expressed as a percentage of the untreated control (100% viability), was calculated for each concentration. Concentration–response curves were built, and the effective cytotoxic concentration 50 of the compounds (CC_50_, μM) was derived. All data points represent an average of three independent assays.

### 3.7. May–Grünwald Giemsa (Pappenheim) Staining

HeLa cells were seeded in a 6-well plate at a density of 2 × 10^5^ cells/well. After incubation for 24 h, the tested compounds were administered at a concentration of 25 µg/mL, and the cells were incubated again for another 24 h. They were fixed with abs. EtOH (30–60 s), stained subsequently with May-Grünwald and 10% Giemsa (in phosphate buffer pH 7.4) and rinsed with water. After vertical drying, the morphological assessment was performed on Zeiss Axio Imager.M2 (Zeiss Group, Oberkochen, Germany).

## 4. Conclusions

The veterinary antibiotic salinomycin reacts with thallium(I) ion under basic conditions to form a mononuclear complex with very asymmetric TlO_6_ coordination, realized by oxygen donor atoms from the pentadentate ligand monoanion and water molecules. Every two coordination species (SalTl1 and SalTl2) are linked to each other by an additional water molecule, achieving the complex composition [TlSal(H_2_O)]×0.5H_2_O. The in vitro antimicrobial and antitumor assays reveal that accommodation of the heavy metal ion in the hydrophilic cavity of the ligand reduces its activity, most likely due to the formation of a stable complex that is unable to dissociate in the intracellular space and disrupt the metal homeostasis of the target bacterial strains and cancer cells. The observed results can serve as a starting point for further investigation of the properties of the polyether ionophore salinomycin under different experimental conditions.

## Data Availability

Data are available from the authors upon request.

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
