# Peer review of "Crystal Structure and Properties of Thallium(I) Salinomycinate"

_ijms, 2025, doi:10.3390/ijms26136504_

Round 1
Reviewer 1 Report
Comments and Suggestions for Authors
This paper entitled "Crystal Structure and Properties of Thallium(I) Salinomycinate" reports on the preparation and characterization of a novel thallium(I) coordination compound [TlSal(H2O)] of the polyether ionophorous antibiotic salinomycin (SalH). The crystal structure is also confirmed. In addition, the effect of thallium(I) binding on the antibacterial and antiproliferative properties of salinomycin was investigated using a series of Gram-positive bacteria and the human tumour HeLa cell line (cervical cancer). The results are original and may be of great interest to the scientific world. As far as I know, the coordination species described here are the second example of a monovalent metal derivative of natural, unmodified salinomycin, besides its sodium analogue, which means that the results obtained fill a gap in this area. The conclusions are consistent with the evidence and arguments presented, and address the main question posed. The references are appropriate.
The figures should be of better quality and clearer, so the authors should increase the resolution.
Why was another cell line not used, especially a healthy cell line?
The English could be improved to more clearly express the research.
Author Response
The answers are uploaded as a file.

Reviewer 2 Report
Comments and Suggestions for Authors
I want to congratulate all the authors on this excellent work. The authors have done an outstanding job characterizing these complex and details analysis on X-ray diffraction analysis. I will be happy to accept this manuscript after minor revisions.
- The authors used various spectroscopic methods to characterize the species. Surprisingly, they did not have any studies in solution, although in one case, they have mentioned “different dissociation degree is important”. I am wondering if they have tried to compare the stability in solution.
- The authors have done a great job using IR and Raman spectroscopy. I would like to see the carboxylate ligand band mentioned in Figures 4 and 5, as they have discussed in the text. Although they have provided a table, it would be great if they only offered zoom spectra from 4000 to 1500 or only the 2000-1500 region. The current figure can go to SI. All the spectra should be overlaid to compare correctly. Now, it’s difficult to analyze from the figure.
- Is it possible to study 13C NMR for these complexes to see the difference in the chemical shift of these carboxylate carbons? This will also provide some solution studies.
- Why is shift in IR stretch for SalH and SalNa or SalK (1704 and 1716/1719) but there is no shift in IR stretch between SalH and SalTl
Author Response
The answers are uploaded.

Reviewer 3 Report
Comments and Suggestions for Authors
The manuscript titled “Crystal Structure and Properties of Thallium(I) Salinomycinate” describes the synthesis and structure of the coordination complex of salinomycin with Tl(I). In addition to the structural description, the IR and Raman spectral characteristics of the title compound were compared with those of salinomycin and its Na- and K-complexes. During antibacterial tests and tests on a cervical cancer cell line, its activity was compared to that of salinomycinic acid, monensic acid, their Na-compounds, and the Tl-monensinate. The motivations and objectives of the work are interesting, and the paper is well-organized. The results and conclusions are presented clearly. However, some corrections would improve the clarity of the manuscript.
In the Results and discussion part, in the first sentence (row 72) “The interaction of salinomycinate anion (SalH, Scheme 1) with thallium(I) nitrate”, the salinomycinate anion is referred as the neutral salinomycinic acid. Earlier in the introduction, the salinomycinate anion is marked as Sal−. It should be marked uniformly throughout the whole text.
The IR and Raman spectra are shown in very small graphs. Some of the presented and discussed characteristics cannot be easily observed on these graphs. For example, bands at 1549 cm−1 and 1398 cm−1 are difficult to locate in Figure 4. Both Figure 4 and Figure 5 should be enlarged (their resolution is high enough), and the specific differences would be highlighted.
Below Figure 6, for B. cereus, the abbreviation BS stands, as for B. subtilis. I suppose that it is misspelled and should stay BC.
Considering all the above-mentioned, I recommend accepting this manuscript for publication with minor revisions.
Author Response
The answers are uploaded as a file.

Reviewer 4 Report
Comments and Suggestions for Authors
This article reports the successful synthesis and structural characterization of a novel thallium(I)-salinomycin complex. Through single-crystal X-ray diffraction and complementary techniques, the authors elucidate the compound’s crystal structure, revealing a unique coordination mode of salinomycin. This work provides a valuable addition to structural and coordination chemistry literature. However, the following aspects require further clarification:
1. The proposed mechanism by which the complex reduces salinomycin’s bioactivity warrants deeper investigation.
2. The influence of thallium ions on salinomycin’s interactions with bacterial/tumor cells remains unclear.
3. Thallium’s status as a toxic heavy metal necessitates a comprehensive discussion of safety implications.
Author Response
The answers are uploaded.

Round 2
Reviewer 4 Report
Comments and Suggestions for Authors
The author has satisfactorily responded to all reviewer comments and has adequately addressed the concerns raised during the review process. Based on the comprehensive revisions and explanations provided, I recommend accepting this manuscript for publication.